# Molecular Pathogenesis of Psoriasis and Biomarkers Reflecting Disease Activity

**DOI:** 10.3390/jcm10153199

**Published:** 2021-07-21

**Authors:** Masaru Honma, Hiroyoshi Nozaki

**Affiliations:** 1Department of Dermatology, Asahikawa Medical University Hospital, 2-1-1-1 Midorigaoka-Higashi, Asahikawa 0788510, Japan; hnozaki@asahikawa-med.ac.jp; 2International Medical Support Center, Asahikawa Medical University Hospital, 2-1-1-1 Midorigaoka-Higashi, Asahikawa 0788510, Japan

**Keywords:** inflammatory skin disease, Th17 cells, adipokines, glycoproteins, fatty acid-binding protein

## Abstract

Psoriasis is a chronic inflammatory skin disease induced by multifactorial causes and is characterized by bothersome, scaly reddish plaques, especially on frequently chafed body parts, such as extensor sites of the extremities. The latest advances in molecular-targeted therapies using biologics or small-molecule inhibitors help to sufficiently treat even the most severe psoriatic symptoms and the extra cutaneous comorbidities of psoriatic arthritis. The excellent clinical effects of these therapies provide a deeper understanding of the impaired quality of life caused by this disease and the detailed molecular mechanism in which the interleukin (IL)-23/IL-17 axis plays an essential role. To establish standardized therapeutic strategies, biomarkers that define deep remission are indispensable. Several molecules, such as cytokines, chemokines, antimicrobial peptides, and proteinase inhibitors, have been recognized as potent biomarker candidates. In particular, blood protein markers that are repeatedly measurable can be extremely useful in daily clinical practice. Herein, we summarize the molecular mechanism of psoriasis, and we describe the functions and induction mechanisms of these biomarker candidates.

## 1. Introduction

Psoriasis is a recurrent, persistent inflammatory skin disorder characterized by rough, reddish plaques on frequently chafed body parts, such as the extensor sites of the extremities [1,2]. Individuals with this condition suffer from subjective symptoms, such as itching and pain, but also from skin lesions, especially on exposed areas, such as the scalp, face, hands, and nails, which can have a prominent impact on the patients’ quality of life [3,4,5,6]. In fact, it has been suggested that the manifestation of psoriasis-related symptoms can trigger stigmatization, leading to social discrimination and alienation [7,8,9]. Psoriasis often coexists with varied comorbidities represented by psoriatic arthritis (PsA), uveitis, psychiatric disorders, metabolic disorders, and cardiovascular diseases [2,10,11,12,13]. Psoriasis is therefore considered to be part of systemic disorders characterized by skin lesions. The process that amplifies localized psoriatic molecular reactions into a systemic inflammatory response is called “psoriatic march” [14].

While severe psoriatic symptoms are often resistant to conventional treatment, recent advances in molecular-targeted therapies have enabled sufficient treatment and control in most cases. The clinical effects of these therapies allow for markedly reduced skin lesions, related symptoms, and comorbidities but also a deep understanding of the molecular mechanism of psoriatic diseases in which the interleukin (IL)-23/IL-17 axis based on Th17-cell-mediating cytokine network plays an essential role [11,15].

Biomarkers are indicators of normal physiological processes, pathogenic reactions, and responses to pathogen/treatment exposure or intervention, including therapeutic interventions [16]. Biomarkers can have molecular biology, histology, radiological images, or other physiologic characteristics [16]. A reliable indicator that reflects sufficient remission of disease activity is indispensable for establishing standardized therapeutic strategies. To date, several biomarker candidates have been proposed to reflect improvement in psoriasis during treatment. Consequently, this review describes the molecular pathogenesis of psoriasis and changes in biomarkers that occur along its disease activity, focusing on blood-protein markers that can be repeatedly measured in daily clinical practice.

## 2. Molecular Pathogenesis of Psoriasis

As shown by the remarkable efficacy of molecular-targeted therapies, the IL-23/IL-17 axis, which depends mainly on Th17-cell function, is considered the most essential mechanism of psoriasis (Figure 1) [1,2,10,15,17]. Molecules regulated in the downstream of Th17 cytokines are identified as biomarker candidates (Table 1). In the initial stage, a complex of antimicrobial peptides (AMP), such as LL-37, and self-nucleotides derived from damaged keratinocytes via toll-like receptors (TLRs), potentiate the function of plasmacytoid dendritic cells (pDCs) to produce substantive interferon (IFN)-α, which activates myeloid (conventional) DCs [18,19,20]. These activated DCs release tumor necrosis factor (TNF) and IL-23 that synergistically propel the immune response process. TNF stimulates DCs in an autocrine manner but inhibits the function of pDCs [1,2,10,15,17]. A paradoxical reaction during treatment using TNF inhibitors can depend on pDC activation by cancelling TNF-mediated inhibition [21].

### 2.1. IL-23

IL-12, IL-23, IL-27, and IL-35 form the IL-12 cytokine family [22] in which subunits and specific receptors are shared [22,23]. For instance, IL-12 and IL-23 share the p40 subunit, but IL-23 specifically possesses the p19 subunit. IL-12 and IL-23 signals are transmitted by pairs of IL-12 receptor β1 (IL-12Rβ1)/IL-12Rβ2 and IL-12Rβ1/IL-23R, respectively. In a psoriatic lesion, both p40 and p19-expressions but also the expression of p40 and p19 subunits increases as opposed to the p35 subunit, which is another component of IL-12, suggesting a more definitive role of IL-23 in the molecular mechanism of psoriasis [24]. IL-12 and IL-23 work differently on T-cell diversity. IL-12 mainly leads to the induction of Th1 cells, whereas IL-23 mainly enhances Th17-cell pathogenicity characterized by IL-17 production [22,23]. IL-23 expression also increases epidermal keratinocytes by TLR-4 stimulation, which can participate in the pathogenesis of interleukin-36 receptor antagonist (DITRA) deficiency [25]. IL-23 stimulates DCs to induce IL-22 release from Th cells [26]. IL-23 expression in keratinocytes is epigenetically regulated, and the mechanism can contribute to the patho-mechanism of psoriasis [27]. IL-23 also potentiates FoxP3-positive regulatory T cells (Treg) to produce IL-17 [28,29]. While DCs are the main source of TNF, TNF is also produced by other cells, such as Th1, Th17, macrophages, neutrophils, mast cells, endothelial cells, and epidermal keratinocytes [11,20,30,31]. TNF induces proinflammatory responses via various signaling pathways, such as nuclear factor (NF)-κB and MAP-kinase signaling, through TNF receptors, which is broadly expressed by various cell types [32]. Consequently, TNF activates DCs and accelerates the inflammatory reactions that involve various immunocytes [10,32]. While IL-23 can fortify the pathogenicity of Th17 cells, it is not required for the differentiation of Th17 cells from naïve CD4+ T cells. In contrast, TGF-β, IL-21, and IL-6 are indispensable for Th17 differentiation, [33,34].

### 2.2. IL-17

IL-17 consists of IL-17A, IL-17B, IL-17C, IL-17D, IL-17E, and IL-17F homodimers or a IL-17A and IL-17F heterodimer. Ligand-specific IL-17 receptors (IL-17R) transmit IL-17 signaling, whereas IL-17A signaling employs IL-17RA, IL-17RC, and IL-17RD [35,36,37,38]. IL-17 receptors share a similar expression of fibroblast growth factor and IL-17R (SEFIR) domain, an intracellular domain essential for recruiting Act-1, a protein that activates NF-κB and MAPK pathways [35,36,37]. IL-17 is indispensable for host defense against cutaneous and mucosal infection by *Staphylococcus aureus* and *Candida albicans* [35,36,37,39] and for upholding the intestinal epithelial barrier [40,41]. IL-17A is the most investigated subtype in both physiological and disease conditions, including psoriasis [36], and it is produced by Th17, Tc17, tissue-resident memory T (T_RM_), innate lymphoid cell (ILC)-3, invariant natural killer T cells (iNKT), gamma delta T cells, and mucosal associated invariant T (MAIT) cells [42]. Free fatty-acid-nourished, CD8-positive T_RM_ is present even in the healed epidermis following psoriasis, and IL-17 released from T_RM_ contributes to lesion recurrence [43,44]. Moreover, IL-17 and IL-22 released from Th17 and other innate cells, such as ILC-3 and gamma delta T cells, induce hyperproliferation of epidermis and accelerate the production of inflammatory cytokines and chemokines, such as IL-8 (CXCL-8), IL-17C, and vascular endothelial growth factor, in the epidermis [1,2,10,17]. According to a study investigating cytokine profile in small and large psoriatic plaques, IL-17 signaling is consistently accelerated in lesional skin; however, suppressed regulation of inflammatory reaction and upregulated TNF signaling are simultaneously observed even in non-lesional skin of patients with large psoriatic plaques. Consequently, this suggests that synergistic effect of IL-17 and TNF signaling induce a systemic inflammatory reaction [45]. These factors play crucial roles in the formation of the psoriatic phenotype and the vicious inflammatory circle of a psoriatic lesion. While the role of IL-17 is broadly shared in other psoriatic diseases, such as PsA [15] and pustular psoriasis [46], the significance of IL-17 inhibition remains unclear in terms of treating palmoplantar pustulosis [47,48].

### 2.3. CCL20/CCR6 Axis

C-C motif chemokine ligand (CCL) 20, a well-known macrophage inflammatory protein (MIP)-3a or liver activation-regulated chemokine, is a member of the CC-chemokine family and plays a significant role in inflammatory and homeostatic conditions [49,50]. Although a constitutively strong expression is observed in the liver, lung, appendix, and tonsils, this expression can be induced in various cells, such as immune, endothelial, and epithelial cells [50]. CCL20 recruits immunocytes expressing the specific receptor CCR6, and CCR6 expression is observed in DCs, T cells, and B cells [50]. The interaction between CCL20 and CCR6 is an indispensable pathogenic mechanism of autoimmune disorders, including psoriasis, and it is considered a distinct therapeutic target [51]. Th17 cytokines and TNF independently and synergistically induce CCL20 expression in epidermal keratinocytes [52,53]. CCL20 expression is significantly upregulated in scratched keratinocyte sheet, suggesting the contribution of CCL20 in the Koebner phenomenon [54]. Deletion of CCR6 or the dominant-negative form of CCL20 ameliorates skin symptoms in psoriasis model mice [55,56,57], thus suggesting the indispensable role of the CCL20/CCR6 axis in the pathogenesis of psoriasis [58,59].

### 2.4. Adipose Tissue

Adipokines and proinflammatory cytokines derived from white adipose tissue (WAT) can enhance and influence the Th17-mediated inflammatory response (Figure 2). Psoriasis is frequently concurrent with obesity and overweight [60,61], which are closely related metabolic abnormalities, and weight reduction interventions are necessary to reduce the severity of skin lesions and comorbidities [62,63,64,65,66,67,68]. Similar to obesity, the expression of proinflammatory adipokines, such as TNF, IL-6, leptin, resistin, and chemerin, is upregulated in psoriasis, whereas the expression of anti-inflammatory adipokines, such as adiponectin and omentin, is suppressed [60,69,70,71,72]. Although they are possibly more closely associated with systemic inflammation, oxidative stress, and cardiovascular risk [73,74], visceral adipose tissue and subcutaneous adipose tissue have similar cytokine profiles [73]. In obese WAT, macrophage infiltration into the stromal vascular fraction of WAT via monocyte chemoattractantprotein (MCP)-1/CCR2 pathway is a key mechanism of obesity-induced adipose inflammation [75]. Adipose tissue macrophages (ATMs), which resemble M1-macrophages, can be activated via TLR4 stimulation by lipopolysaccharide and saturated fatty acids (SFAs) [76] and release proinflammatory cytokines, such as TNF and IL-6 [77]. SFAs, pathogen-associated molecular patterns (PAMPs), and danger-associated molecular patterns (DAMPs) also activate NLRP3 inflammasomes in ATMs, resulting in enhanced production of IL-1 and IL-18 [78]. WAT also acts as a reservoir of T_RM_ cells that is characterized by high turnover rates and active metabolism, as measured by lipid uptake and mitochondrial respiration [79,80]. The numbers of CD8+ T_RM_ cells can be present in psoriatic skin for long periods, taking in free fatty acids via fatty acid-binding protein (FABP)-4/5 for the regional longevity [44]. These cells play a crucial role in the recurrence of clinically healed psoriasis [43].

## 3. Biomarkers in Psoriasis Treatment

### 3.1. Peripheral Blood Cell Counts

#### Neutrophil-to-Lymphocyte Ratio and Platelet-to-Lymphocyte Ratio

Neutrophils and platelets are primarily associated with biophylactic mechanisms against pathogens and hemostasis, respectively. These mechanisms synergistically work at sites of acute injury and inflammation by forming neutrophil extracellular traps. Dysregulated interaction between neutrophils and platelets can be involved in the patho-mechanism of autoimmune disorders, such as systemic lupus erythematosus (SLE), rheumatoid arthritis (RA), systemic vasculitis [81], and psoriasis [82].

Recently, neutrophil-to-lymphocyte ratio (NLR) and platelet-to-lymphocyte ratio (PLR) have been considered as markers of systemic inflammation in internal malignancies [83] and various inflammatory conditions, such as SLE and RA [84]. While systemic treatment using biologics can reduce NLR and PLR and improve psoriatic skin and arthropathic symptoms [85,86], it is not always correlated with the severity of psoriasis skin lesions as evaluated by the Psoriasis Area and Severity Index (PASI), suggesting that the NLR and PLR are better at reflecting systemic inflammation [87].

### 3.2. Cytokines and Chemokines

#### 3.2.1. IL-17

As mentioned earlier, IL-17 is a definitive mediator in the patho-mechanism of psoriasis, and it is the most important subtype, as shown by the excellent clinical efficacy of the inhibitors against psoriasis.

Serum IL-17 levels increase as the severity of skin lesions increases, especially in severe psoriatic cases [88], and IL-17A levels are more closely correlated with psoriasis severity compared to IL-17F levels [89]. In contrast, serum IL-23 levels do not increase in psoriatic cases, and changes cannot be detected during successful treatment [90]. IL-17A and IL-17F are targets of IL-17-specific inhibitors but also of other drugs. Tofacitinib, a JAK-inhibitor, and apremilast, a phosphodiesterase inhibitor, decrease serum IL-17A, and IL-17F levels are correlated with the clinical response of skin lesions [91,92]. Serum levels of both subtypes change over the course of treatment and the withdrawal of guselkumab, an IL-23 p19-specific inhibitor [90]. Interestingly, increased serum levels of IL-17F subunit precede skin lesion exacerbation after withdrawal of guselkumab therapy [90], which might depend on the sensitivity of measuring these subunits. While IL-17A and IL-17F are mainly produced by immune cells, such as Th17 and Tc17 cells, the latter is also produced by colon epithelial cells [93], and serum IL-17F levels are significantly higher than serum IL-17A levels [89,90]. IL-17A is also related to the progression of cardiovascular disease, fatty liver, and diabetes [94,95,96]. Consequently, IL-17A-inhibition can possibly improve non-calcified atherosclerosis of coronary arteries [97].

#### 3.2.2. IL-22

IL-22 is a member of the IL-20 subfamily of cytokines, which belong to the IL-10 family, and it is produced by Th17, Th22, ILC3, Tc22, and gamma delta T cells. However, it plays a crucial role in tissue regeneration, wound healing, and host defenses, especially against fungal infections [98,99,100]. The signal can be transmitted via a pair of receptors (IL-10 and IL-22) through JAK/STAT pathways [15,99,100]. IL-22 upregulates the proliferation of epidermal keratinocytes and induces acanthosis of epidermis via STAT3 activation in inflammatory dermatoses, such as psoriasis and atopic dermatitis (AD) [101,102]. Serum IL-22 levels increase moderately in psoriasis, in accordance with the skin lesion severity as evaluated by the PASI score [90,103], whereas these levels decrease when implementing an appropriate treatment [90,104]. IL-19, another subfamily of the IL-10 family produced by monocytes, macrophages, keratinocytes, and fibroblasts, is involved in inflammation, angiogenesis, and tissue remodeling [98,105]. Serum IL-19 levels increase in cases of plaque-type psoriasis, and they are very closely correlated with the skin lesion severity as rated by the PASI score [106]. Elevated IL-19 levels in psoriasis can quickly be reduced by ixekizumab or baricitinib treatment. The therapeutic response of psoriasis is predicted by the decrease in the serum IL-19 levels before skin lesions begin to heal [106]. The IL-20 family is also associated with other systemic diseases. IL-19 and IL-22 can be vascular protective cytokines in cardiovascular diseases [107], whereas the synergistic effect of IL-22 and IL-17A can contribute to fibrotic changes in the liver tissue [108].

#### 3.2.3. IL-36

IL-36, an IL-1 family proinflammatory cytokine, consists of IL-36α, IL-36β, and IL-36γ. The IL-36 signal induces an inflammatory response in various tissues [109,110,111,112]. IL-36 family of cytokines are produced by immune cells, such as macrophages, DCs, and T cells but also by epithelial tissues, including the epidermis [109,113,114,115]. Among its subtypes, IL-36α and IL-36γ are significantly expressed in psoriatic epidermis, and the expression can be induced by proinflammatory cytokines that are deeply involved in the molecular patho-mechanism of psoriasis, such as IL-17 and TNF [116]. Furthermore, IL-36 and IL-17A synergistically propel a vicious inflammatory loop [113,114,117]. Serum IL-36γ levels are increased in cases of plaque-type psoriasis, and they are closely correlated with the respective severity; however, the elevated levels can be normalized when adequate treatment is provided [118]. Elevated serum IL-36γ levels constitute a relatively specific diagnostic marker for psoriatic erythroderma that is differentiated from other erythrodermic dermatoses [119].

#### 3.2.4. Fractalkine

Fractalkine (CX3CL1) is a CX3C chemokine expressed in antigen-presenting cells [120], vascular endothelial cells [121], and epidermal keratinocytes [122] in membrane-bound or soluble forms. Fractalkine works as an inflammatory mediator via the specific CX3C chemokine receptor 1 (CX3CR1), and fractalkine expression increases in lesional psoriatic epidermis [122]. This elevated expression contributes to the recruitment of CXCR1-expressing cells, such as natural killer cells, T cells, and monocytes, via the chemotactic effect of the soluble form [123]. Experiments on CX3C-deleted mice revealed that imiquimod could attenuate psoriasis-like inflammation, thus suggesting a key role of the fractalkine/CX3CR1 signaling in the pathogenesis of psoriasis [124]. Serum fractalkine levels increase in cases of psoriasis and AD depending on skin lesion severity [125,126]. Although elevated serum fractalkine levels decrease along with improvement of AD, there are no data on serum fractalkine level changes during psoriasis treatment. Fractalkine is also involved in the molecular mechanism of atherosclerosis [127], and its expression can reflect a systemic inflammatory reaction.

#### 3.2.5. Thymus and Activation-Regulated Chemokine

Thymus and activation-regulated chemokine (TARC)/CCL17 is one of the CC chemokines expressed in the thymus and is produced by various cells, such as dendritic cells (DC), endothelial cells, keratinocytes (KC), bronchial epithelial cells, and fibroblasts [128,129]. The signal is transmitted by the specific receptor CCR4, resulting in lesional infiltration of Th2 cells, basophils, and natural killer cells [129]. TARC is one of the most useful biomarkers for reflecting the current disease activity of AD. TARC expression is slightly upregulated in lesional psoriatic skin, and numbers of CCR4-expressing mononuclear cells infiltrate the lesional skin, suggesting the possible involvement of TARC in the patho-mechanism of psoriasis [130]. While serum TARC levels are lower in psoriasis cases compared to AD cases [131], they tend to increase in more severe cases of psoriasis [132]. Interestingly, the serum TARC level also increases in well-controlled psoriasis cases treated with biologics, especially IL-17 inhibitors [132]. The ILC2 population can possess ILC3-like characteristics when IL-1β and IL-23 are stimulated, both of which are pivotal cytokines in the psoriatic molecular pathogenesis [133]. Details of the induction mechanism of TARC remain unclear, but this process may involve the plasticity of immune cells. In addition, serum levels are higher in cases of generalized pustular psoriasis compared to cases of plaque-type psoriasis, suggesting a relationship with psoriasis severity [134].

### 3.3. Adipokines

Adipokines (or adipocytokines) are adipose, tissue-derived bioactive proteins that play an essential role in regulating tissue metabolism. Depending on their physiological and pathological effects, adipokines can be classified into proinflammatory and anti-inflammatory groups [135]. In obesity, the balance of adipokines will shift toward a dominant condition of proinflammatory adipokines, and the aberrant secretion contributes to latent systemic inflammation [72,135,136,137]. These abnormal adipokine states are shared by obesity and psoriatic diseases in which the expression of proinflammatory adipokines leptin, resistin, and chemerin increases, as opposed to the expression of anti-inflammatory adipokines, i.e., adiponectin and omentin, which decreases [69,70,71,138]. Leptin, which can regulate feeding behaviors by acting on the central nervous system, induces the production of TNF, IL-6, and CC-chemokine from monocytes and macrophages as well as IL-2 and IFN-γ from T cells [135]. Among these adipokines, chemerin, lipocalin-2, resistin, and adiponectin are better biomarker candidates for reflecting psoriasis severity [139].

#### 3.3.1. Resistin

Initially identified in adipose tissue, resitin can be produced in greater quantities by macrophages and monocytes in humans, and its expression is induced by proinflammatory cytokines, such as TNF, IL-1, and IL-6 [135]. Serum resistin levels accurately reflect insulin resistance, and resistin inhibition partially improves the aberrant insulin function [140]. Resistin signaling upregulates the production of proinflammatory cytokines from mononuclear cells, thus forming a vicious inflammatory circle [135]. Plasma resistin levels are correlated with the severity of psoriatic skin lesions, and its levels can decrease as the skin lesions improve following an appropriate treatment approach [141,142]. While serum resistin levels are closely correlated to the PASI score and to the involved body surface area percentage (%BSA) in psoriasis cases before anti-TNF therapy, its levels do not always decrease with the improvement in PASI and %BSA after adalimumab therapy [143]. Serum resistin and leptin levels are also correlated with the intima-media thickness of carotid arteries in psoriasis cases, suggesting their potential contribution to the development of atherosclerosis [144].

#### 3.3.2. Adiponectin

Adiponectin enhances insulin-sensitivity but reduces the TNF-induced dysfunction of endothelial cells and apoptosis of cardiomyocytes [145]. Adiponectin mitigates imiquimod-induced psoriasiform dermatitis via the direct inhibition of IL-17 release from gamma delta T cells [146]. Furthermore, serum adiponectin levels are inversely correlated with skin lesion severity [147,148], and its levels do not always increase with the improvement in skin lesions [142,149]. Serum adiponectin levels exhibit a greater decrease in cases with PsA compared to cases without PsA, suggesting a closer relationship between adiponectin and systemic inflammatory responses [150].

### 3.4. Antimicrobial Peptides

AMPs are small proteins with approximately 10–50 amino acids that demonstrate biophylactic activity against viral, bacterial, and fungal infections via the disruption of the pathogens’ plasma membrane. The main cellular sources for AMPs in the human skin are keratinocytes, mast cells, neutrophils, sebocytes, and eccrine epithelial cells [19,151,152,153]. AMPs play a critical role in innate immunity, and they are involved in chemotaxis, angiogenesis, and cell proliferation/migration during the host’s inflammatory responses [154]. AMP expression is highly upregulated in psoriatic epidermis and is possibly involved in the patho-mechanism of psoriasis [19,154].

#### 3.4.1. Defensin 2

β-defensin 2 (BD-2), a defensin subfamily, is the most investigated molecular biomarker of psoriasis. BD-2 expression is induced by proinflammatory cytokines and microbial products in contrast to the constitutive expression of BD-1 in epithelial cells [155]. TNF, IFN-γ, and IL-17, which are closely involved in the pathogenesis of psoriasis [154], can induce the BD-2 expression in epidermal keratinocytes, and TNF and IL-17A synergistically enhance BD-2-induction [53]. In cases of plaque-type psoriasis, BD-2-protein levels significantly increase both in lesional epidermis and in serum, and serum levels are closely correlated with skin lesion severity as rated by the PASI score [156] and with serum IL-17A levels but not with the IL-17F levels [89]. Several clinical trials have evaluated the efficacy of novel therapeutic options for psoriasis by measuring BD-2 levels [89,157,158]. In moderate to severe psoriasis, elevated serum BD-2 levels decreased and were normalized as the PASI score improved [89,157,158].

#### 3.4.2. S100A

S100 proteins (measuring 10–12 kilodaltons) are low molecular-weight molecules that possess two calcium-binding helix-loop-helix motifs, and they form a family that consists of 25 subtypes [159]. Although S100A7 (psoriasin), S100A8, S100A9, S100A12, and S100A15 (koebnerisin) exhibit antimicrobial activity and are highly expressed in psoriatic epidermis [159,160], S100A7 is the most studied subtype. Proinflammatory cytokines deeply involved in the pathogenesis of psoriasis, such as IL-36, IL-17, and TNF, can independently and synergistically induce S100A7 expression in epidermal keratinocytes, and S100A7 acts as a chemoattractant for lymphocytes, granulocytes, and macrophages, forming an inflammatory loop [161]. Serum S100A7 levels increase in severe psoriatic cases but not in milder ones [162]. Serum S100A7 and S100A15 levels are closely correlated with the intima-media thickness of common carotid arteries [163], suggesting their contribution to the systemic inflammatory response [164].

### 3.5. Protease Inhibitors

#### 3.5.1. Squamous Cell Carcinoma Antigen

Squamous cell carcinoma antigen (SCCA), which is a recognized serum tumor marker for SCC, is a member of the serpin family of proteins with inhibitory activity against cysteine protease. While SCCA is composed of SCCA1 (SERPINB3) and SCCA2 (SERPINB4), both subtypes are expressed in psoriatic epidermis [165]. SCCA2 expression is significantly upregulated in psoriatic epidermis compared with the normal epidermis in contrast to the constitutive SCCA1 expression in normal and psoriatic epidermis [165]. In psoriasis cases, serum SCCA2 levels are correlated with the PASI score and serum IL-22 levels but not with the IL-17A levels [166]. IL-22, which is involved in the mechanisms of psoriasis and AD, stimulates SCCA1/2 expression in oral SCC-derived cell lines [167] and normal human keratinocytes [166]. IL-17 synergistically acts on the IL-22-mediated induction of SCCA2 in normal keratinocytes [166]. IL-4 and IL-13 signaling can also induce SCCA2 expression in keratinocytes [168]. Thus, serum SCCA2 levels increase in psoriasis but also in other inflammatory dermatoses, such as AD [166,169]. The increased serum SCCA2 levels in psoriasis and AD can be reduced with appropriate treatment [166,169].

#### 3.5.2. Elafin

Elafin, a serine protease inhibitor that is highly expressed in psoriatic epidermis [170,171,172], is released by epithelial cells and immune cells [173] and plays an essential role in the anti-inflammation mechanism via proteinase inhibition and antimicrobial/immunoregulatory functions [173]. Serum elafin levels increase in psoriasis cases correlate with skin lesion severity and with laboratory findings that reflect inflammation, such as C-reactive protein levels and erythrocyte sedimentation rates [174]. During a cardiovascular event, elafin possibly reduces tissue injury exacerbated by neutrophilic elastase as a result of anti-inflammatory activity [175]. Interestingly, higher elafin expression is associated with a higher likelihood of spontaneous reperfusion, and it is related to a smaller infarct size and more favorable clinical outcomes [176].

### 3.6. Glycoproteins

#### 3.6.1. Leucin-Rich Alpha-2-Glycoprotein

Leucin-rich alpha-2-glycoprotein (LRG), an approximately 50 kilodalton glycoprotein consisting of abundant amino acid residues with a structure of leucine-rich repeats (LRP), is produced by hepatocytes, neutrophils, endothelial cells, and macrophages following the stimulation of proinflammatory cytokines, such as IL-6, TNF, and IL-1β. LRG is associated with angiogenesis in cooperation with TGF-β signaling [177], and serum LRG levels are a candidate biomarker that reflects cardiovascular risk in cases of kidney diseases [178]. LGR has also been involved in a Th17-differentiation mechanism in a collagen-induced arthritis model [179]. While serum LRG levels increase in cases of psoriasis, depending on the skin lesion severity, its levels are much more closely correlated with serum C-reactive protein levels than with the PASI score [180]. Considering that serum LRG levels are higher in psoriatic cases with arthritis than in cases without arthritis, serum LRG levels might be more reflective of a systemic inflammatory response than of the skin-limited inflammatory level [180].

#### 3.6.2. YKL-40

Chitinase-3 -like 1, also known as YKL-40, is a glycoprotein that contains highly conserved chitin-binding domains without chitinase activity [181,182,183]. YKL-40 is secreted by various immune cells, such as neutrophils and macrophages, fibroblasts, vascular smooth muscle cells, and endothelial cells [181,182,183]. YKL-40 expression is upregulated by proinflammatory cytokines, namely IL-6, TNF, IL-13, and IL-18, and is associated with tumor progression, angiogenesis, and various inflammatory responses [181,182,183]. In psoriatic lesions, YKL-40 expression is detected in infiltrating neutrophils, and serum YKL-40 levels are significantly more elevated in cases of generalized pustular psoriasis compared to cases of plaque-type psoriasis [184]. The levels are moderately correlated with skin lesion severity, and they can be reduced following an appropriate treatment [184,185]. Serum YKL-40 levels are also correlated with arthritis and endothelial dysfunction in cases of psoriasis [186,187], suggesting a close correlation with the systemic inflammatory response.

### 3.7. Fatty Acid-Binding Protein

The fatty acid-binding protein (FABP) family includes several tissue-specific subtypes of FABP that exhibit prominent affinity with long-chain fatty acid and play a significant role in lipid metabolism [188,189,190]. Among them, FABP-5 (epidermal FABP, psoriasis-associated-FABP) is highly expressed in psoriatic as opposed to healthy epidermis [191,192,193], and FABP-5 regulates the differentiation of epidermal keratinocytes [194,195]. There have been numerous studies suggesting a close correlation among blood FABP-4 levels, an adipocyte subtype, and metabolic abnormality related to cardiovascular diseases [188]. FABP-4 and FABP-5 are also specifically expressed in T_RM_ cells compared with other T-cell subtypes, and T_RM_ cells require lipid uptake via FABP-5 and FABP-5 to maintain their longevity in the targeting tissues, such as in psoriatic lesional epidermis [44]. Alteration of the blood fatty acid profile in psoriasis also suggests an essential role for FABP in the pathogenesis of this condition [196]. While FABP-4 does not always relate to psoriasis severity, the serum level increases in psoriasis cases compared with healthy controls and decreases with appropriate treatment [197]. Serum FABP-4 levels are inversely correlated with serum TARC levels, which is possibly related to psoriasis remission [198,199]. Moreover, serum FABP-1 (liver-FABP) levels increase in cases of psoriasis depending on skin lesion severity [200], and FABP-2 (intestinal FABP) potentially reflects the subclinical disruption of the intestinal barrier in severe psoriasis cases [201].

## 4. Conclusions

The novel and highly efficient therapeutic approaches in psoriasis have enabled the treatment of recalcitrant psoriatic lesions and comorbidities, thus leading to disease remission. The excellent efficacy of molecular-targeted therapies also highlights and reflects the molecular pathogenesis of psoriatic diseases. To refine the underlying therapeutic strategy, useful biomarkers that can reflect disease severity and sufficient remission are indispensable. Further basic and clinical research is required to establish an optimized therapeutic strategy in psoriasis treatment.

## Figures and Tables

**Figure 1 jcm-10-03199-f001:**
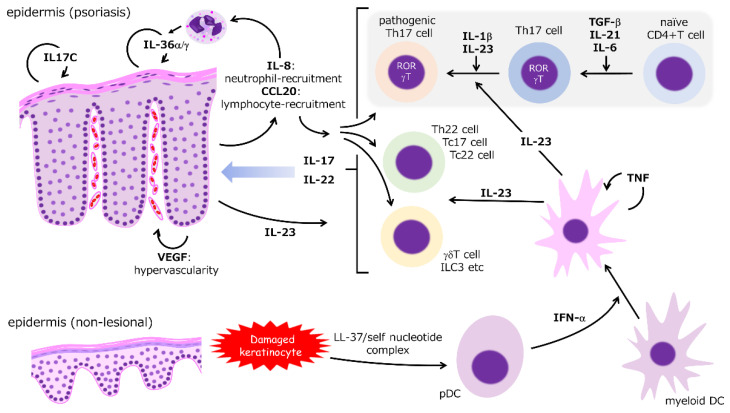
Summarized molecular mechanism of psoriasis. IFN-α released from activated pDCs stimulates myeloid DCs to produce TNF and IL-23. TNF activates DCs in an autocrine manner and enhances the inflammatory responses of various immunocytes. Naïve CD4-positive T cells differentiate into Th17 cells in the presence of the transforming growth factor (TGF)-β, IL-21, and IL-6. The pathogenicity of Th17 cells is potentiated by IL-23. IL-17 and IL-22 are produced by Th17 and other cells with more innate characteristics (e.g., innate lymphoid cell (ILC)-3 and gamma delta T cells). IL-17 and Il-22 induce epidermal hyperproliferation. IL-17 and TNF synergistically accelerate the production of inflammatory cytokines and chemokines from the epidermal keratinocytes, resulting in a vicious circle of inflammatory reactions.

**Figure 2 jcm-10-03199-f002:**
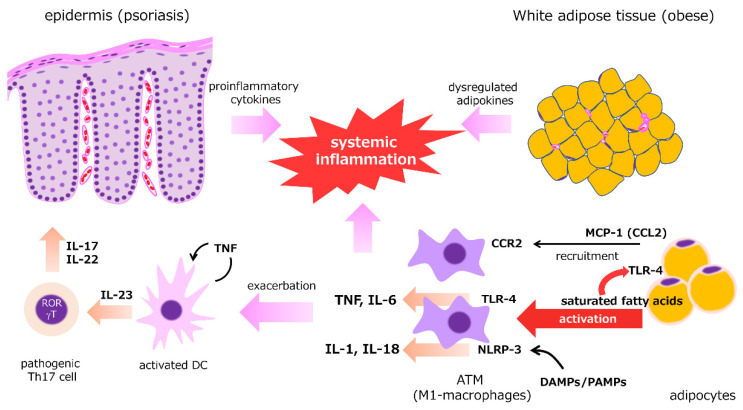
Close correlation between psoriasis and adipose tissue. Adipokines and proinflammatory cytokines derived from white adipose tissue (WAT) enhance and influence the Th17-mediated inflammatory response. In psoriasis and obesity, balance between proinflammatory adipokines and anti-inflammatory adipokines is dysregulated. In obese WAT, macrophages infiltrate the stromal vascular fraction of WAT via the monocyte chemoattractantprotein-1 (MCP-1)/CCR2 pathway. Adipose tissue macrophages (ATMs) activated via TLR4 stimulation by saturated fatty acids (SFAs) release proinflammatory cytokines, such as TNF and IL-6. SFAs, pathogen-associated molecular patterns (PAMPs), and danger-associated molecular patterns (DAMPs) also activate NLRP3 inflammasomes in ATMs, resulting in the enhanced production of IL-1 and IL-18. These proinflammatory cytokines synergistically work with Th17-derived cytokines to enhance systemic inflammatory responses.

**Table 1 jcm-10-03199-t001:** Summary of blood-protein biomarkers reflecting the severity of psoriasis.

Group	Biomarkers	Cellular Source	Findings
blood cell counts	NLR	-	increase especially in cases with arthritis
	PLR	-	
cytokines	IL-17A	Th17, Tc17, ILC3, etc.	relation with atherosclerosis, fatty liver, and insulin resistance
	IL-17F	Th17, Tc17, ILC3, colon epithelial cells, etc.	much higher serum IL-17F levels than IL-17A levels
	IL-22	Th17, Th22, Tc22, ILC3, etc.	vascular protective effect; relation with liver fibrosis
	IL-19	monocytes, macrophages, keratinocytes, fibroblasts, etc.	vascular protective effect
	IL-36γ	epidermis	relatively specific to skin lesions
chemokines	Fractalkine	APCs, ECs, and epidermis	close correlation with atherosclerosis
	TARC	DCs, ECs, epidermis, and fibroblasts	a biomarker for AD; possible relation to deeper remission during anti-IL-17 therapy; correlation with severity of GPP
adipokines	Resistin	macrophages, monocytes, and adipocytes	close correlation with atherosclerosis
	Adiponectin	adipocytes	negatively correlated with atherosclerosis
AMPs	β-defensin 2	epidermis	relatively specific to skin lesion;
	S100A7	epidermis	correlation with atherosclerosis
protease inhibitors	SCCA2	epidermis	also increase in AD
	Elafin	Epidermis and immune cells	correlation with CRP and ESR
glycoproteins	LRG	hepatocytes, neutrophils, ECs, and macrophages	correlation with CRP and arthritis
	YKL-40	neutrophils, macrophages, fibroblasts, ECs, and smooth muscle cells	correlation with tumor progression, metabolic diseases, and arthritis
FABPs	FABP-4	adipocytes	increase in cardiovascular diseases; the expression in T_RM_ infiltrating into psoriatic epidermis
	i-FABP	intestine epithelial cells	correlation with disruption of intestine barrier

AMP, antimicrobial peptide; FABP, fatty acid-binding protein; NLR, neutrophils/lymphocytes ratio; PLR, platelets/lymphocytes ratio; TARC, thymus and activation-regulated chemokine; LRG, leucin-rich alpha-2 glycoprotein; APCs, antigen presenting cells; ECs, endothelial cells; DCs, dendritic cells; AD, atopic dermatitis; GPP, generalized pustular psoriasis.

## Data Availability

Not applicable.

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
