# Peer review of "Molecular Pathogenesis of Psoriasis and Biomarkers Reflecting Disease Activity"

_jcm, 2021, doi:10.3390/jcm10153199_

Round 1

Reviewer 1 Report

The authors if this review article aimed to describe the molecular mechanism of psoriasis and the blood protein biomarkers useful for establishing standardized therapeutic strategies, based on the ability to induce sufficiently deep remissions of the disease.

Despite the title and the programmatic intent, this article does not really provide a useful update on the molecular mechanism of psoriasis. As a matter of fact, the article turns out to be an acritical compilation of biomarkers, whose relationships with the targeted molecular therapies which may have suggested their exploitation also remain largely unexplored.

After an expeditious introduction, section 2, entitled “Molecular mechanisms of psoriasis”, actually starts with an inventory of blood protein markers reflecting the severity of psoriasis, which are not discussed in detail and should have been placed more appropriately in section 3 (“biomarkers in psoriasis treatment”). The paragraphs in section 2 aiming at a description of the pathogenic mechanisms in psoriasis, i.e.., paragraph 2.1 (“IL-23”), 2.2 (“IL-17”), 2.3 (“CCL20/CCR6 axis”) and 2.4 (“Adipose tissue”) are characterized by a general lack of clarity. In all of them, large amounts of unnecessary generalities are provided together with a number of particular observations, but the authors fail to show which mechanistic links may be connecting them with one another. Issues of primary pathogenic importance, such as the dysregulation of epidermal cell proliferation, the induction of autoimmune responses, the maintenance of immunity-driven inflammation, the role of oxidative stress (and lipid peroxidation, I may add) are mentioned, but not really addressed. In paragraph 2.4, no sufficiently clear explanation is provided of the possible relationships linking psoriasis with the hormones produced by adipose tissue and the cytokines produced by adipose tissue-associated lymphoid cells.

Section 3 is a list of biomarkers, with a description of the correlations reported with disease activity and/or the response to therapies. However, a critical discussion is lacking of their usefulness, reliability and appropriateness in relation with the clinical situation, which should be considered most needed by clinicians, as no one of them would be wanting to try the entire list of possible biomarkers in any given clinical setting.

The article is in need for a general linguistic revision.

Author Response

I appreciate the essential comments on this manuscript. As the reviewer pointed out, this field has not been sufficiently clarified. Therefore, I try to summarize the findings about biomarkers on the psoriatic treatment. In this version, the descriptions have been proofread by a native specialist again, and many parts have been linguistically amended. In addition, the figures are replaced by new figures with higher resolution. Changed descriptions can be checked using the “Track changes” function of MS WORD.

Thank you very much for the valuable comments again.

Reviewer 2 Report

The authors presented a lot of interesting data on the biomarkers of psoriasis severity. The article is well written and designed. It is a pleasure to read. The structure of the article makes it easier for the reader to get acquainted with the topic. The authors summarized the current knowledge on biomarkers in psoriasis and documented the theses presented with numerous references, which is an additional strength of the article. I believe that apart from minor editorial corrections, the article does not require any corrections and can be published in its current form.
Minor comments:
In Table 1, the word apiponectin was used - please correct.
The presented figures are very interesting, but it would be better if they were of better resolution.

Author Response

Thank you very much for the encouraging comment.

The misspelling and the problem on the figures have been checked and corrected. Changed descriptions can be checked using the “Track changes” function of MS WORD.

Thank you very much for the valuable comments again.

Reviewer 3 Report

In the manuscript „Molecular mechanism of psoriasis and the biomarkers reflecting disease activity” the authors provide a review of great detail about key regulators and drivers of the autoimmune disease psoriasis. In the first part of the review the molecular mechanisms of this disease are enlightened and are supported by clear schemes and tables. Unfortunately, these schemes are of low quality and need to be approved before publishing. In the second part a multitude of relevant biomarkers are listed. I really appreciate that the role of all biomarkers is also set into context of comorbidities. What I am missing in the second part are most recent discoveries such as miRNA levels, which are crucial regulators of gene expression and drive specific gene expression profiles. For instance, the expression of miR-143 and miR-223 in peripheral blood mononuclear cells is positively related with psoriasis disease severity index  (Løvendorf et al., 2014). Additionally, miR-146b inhibits keratinocyte proliferation and miR-203 reverses IL-17 signalling, leading to the conclusion that miRNAs are not only important biomarkers but can only considered as therapeutic targets (Hou et al., 2016; Xu et al., 2017).

In my opinion this review constitutes added value to the field and is ready for publication after minor check spellings.   

Author Response

Thank you very much for the encouraging comment.

Unfortunately, these schemes are of low quality and need to be approved before publishing.

  • I apologize the image quality. The figures have been replaced by novel figures with higher resolution.

What I am missing in the second part are most recent discoveries such as miRNA levels, which are crucial regulators of gene expression and drive specific gene expression profiles.

  • I strongly agree with the reviewer’s comment. MiRNAs are other important biomarkers reflecting psoriasis disease activity. However, there have been numbers of reports describing usefulness of miRNA as biomarkers of psoriasis. Therefore, I decide to eliminate the findings in this manuscript, avoiding information overload. These information about miRNAs should be reviewed at another opportunity.

Thank you very much for the valuable comments again.

Round 2

Reviewer 1 Report

The authors of this manuscript did not deem it necessary to consider any of my major comments or to reform their manuscript in accordance with them. The only innovations introduced in the revised version of the manuscript include a general linguistic revision and the expansion of two figures. From my point of view, I do not find sufficient reasons to change my opinion on this paper, which therefore I continue to consider not worthy of publication in the Journal of Clinical Medicine.

Author Response

I appreciate the essential comments on this manuscript. While we try to summarize the essential findings about biomarkers on the psoriatic treatment, at this moment, it is difficult to response the requests from the reviewer completely. 

Thank you very much for the valuable comments again.